# Evaluation of Different Compatibilization Strategies to Improve the Performance of Injection-Molded Green Composite Pieces Made of Polylactide Reinforced with Short Flaxseed Fibers

**DOI:** 10.3390/polym12040821

**Published:** 2020-04-04

**Authors:** Ángel Agüero, David Garcia-Sanoguera, Diego Lascano, Sandra Rojas-Lema, Juan Ivorra-Martinez, Octavio Fenollar, Sergio Torres-Giner

**Affiliations:** 1Technological Institute of Materials (ITM), Universitat Politècnica de València (UPV), Plaza Ferrándiz y Carbonell 1, 03801 Alcoy, Spain; anagrod@epsa.upv.es (Á.A.); dagarsa@dimm.upv.es (D.G.-S.); juanivorra@outlook.com (J.I.-M.); ocfegi@epsa.upv.es (O.F.); 2Escuela Politécnica Nacional, 17-01-2759 Quito, Ecuador; 3Novel Materials and Nanotechnology Group, Institute of Agrochemistry and Food Technology (IATA), Spanish National Research Council (CSIC), Calle Catedrático Agustín Escardino Benlloch 7, 46980 Paterna, Spain

**Keywords:** PLA, flax, green composites, fiber pretreatment, reactive extrusion

## Abstract

Green composites made of polylactide (PLA) and short flaxseed fibers (FFs) at 20 wt % were successfully compounded by twin-screw extrusion (TSE) and subsequently shaped into pieces by injection molding. The linen waste derived FFs were subjected to an alkalization pretreatment to remove impurities, improve the fiber surface quality, and make the fibers more hydrophobic. The alkali-pretreated FFs successfully reinforced PLA, leading to green composite pieces with higher mechanical strength. However, the pieces also showed lower ductility and toughness and the lignocellulosic fibers easily detached during fracture due to the absence or low interfacial adhesion with the biopolyester matrix. Therefore, four different compatibilization strategies were carried out to enhance the fiber–matrix interfacial adhesion. These routes consisted on the silanization of the alkalized FFs with a glycidyl silane, namely (3-glycidyloxypropyl) trimethoxysilane (GPTMS), and the reactive extrusion (REX) with three compatibilizers, namely a multi-functional epoxy-based styrene-acrylic oligomer (ESAO), a random copolymer of poly(styrene-*co*-glycidyl methacrylate) (PS-*co*-GMA), and maleinized linseed oil (MLO). The results showed that all the here-tested compatibilizers improved mechanical strength, ductility, and toughness as well as the thermal stability and thermomechanical properties of the green composite pieces. The highest interfacial adhesion was observed in the green composite pieces containing the silanized fibers. Interestingly, PS-*co*-GMA and, more intensely, ESAO yielded the pieces with the highest mechanical performance due to the higher reactivity of these additives with both composite components and their chain-extension action, whereas MLO led to the most ductile pieces due to its secondary role as plasticizer for PLA.

## 1. Introduction

In the field of polymers technology, one of the most sought and researched objectives nowadays is the replacement of petroleum derived materials with natural and renewable ones. Moreover, international environmental legislation and modern society increasingly require the manufacture of materials and end products that follow a circular model by design [1,2]. Polymer composites are usually a recurrent solution in materials engineering due to its low weight, easy manufacturing, and high mechanical strength [3,4]. The current trend in polymer composites is focused on the use of biopolymers and natural fillers to develop the so-called “green composites” [5]. Over the last years, a wide range of biodegradable polymers derived from both natural and petrochemical sources have been studied, which undergo hydrolytic or enzymatic decomposition by interaction with microorganisms in compost conditions [6,7]. Among them, polylactide (PLA) has been widely employed as the green composite matrix and it is currently considered the front-runner in the bioplastics market with an annual production of approximately 140,000 tons [8].

Fillers have been widely used in polymer composites [9,10]. Natural fillers can be derived from minerals, animals, or plants. In particular, plants derived fillers, both particles and fibers, can find a wide range of applications in green composites due to their abundance and low cost [11,12]. They can also increase the biodesintegration rate of biodegradable polymers in soil due to an increase in active surface area available for the action by degrading microorganisms [13]. Moreover, they can be obtained from agricultural and industrial wastes or food processing by-products, allowing the valorization of discarded materials [14]. Consequently, several green composites have been recently developed using a wide variety of fillers derived from plants and trees such as pineapple leaf fibers [15], babassu nut shell flour [16], jute fabric [17], banana stem fiber [18], orange peel [19], coconut fibers [20], rice husk [21], almond shell [22] walnut shell [23], etc. It is evident that each type of plant has a specific chemical composition and structure. However, it is also worth noting that every elementary vegetable fiber consists on a single plant cell that has a common microstructure. This structure is based on oriented highly crystalline cellulose microfibrils embedded in a matrix formed by lignin and amorphous hemicellulose, which is organized around an empty space denominated “lumen” [24,25]. These two differentiated cell parts, that is, lignin and hemicellulose (matrix) and cellulose microfibrils (reinforcement), result in different behaviors. While the amorphous matrix (primary wall) keeps the fiber integrity, the oriented microfibrils (secondary wall) receive and transfer the mechanical stresses [26]. In fact, the orientation of the microfibrils highly determines the physical and mechanical properties of the fiber [27]. The microfibrils angle is specific of each type of plant and there may be slight differences among fibers belonging to the same plant as well as the cellulose, lignin, and hemicellulose contents [28]. Hence, due to these microstructural variability, vegetable fibers associate some disadvantages and they rarely fulfill the potential expected in the polymer composites or, even, their behavior as reinforcement shows uncertainty under short- and long-term loadings [29]. Furthermore, they habitually present low thermal stability, showing degradation when subjected to temperatures above 200–250 °C and limiting the processing of the composites by melting routes such as extrusion or injection molding [30].

Linen (*Linum usitatissimum*) is traditionally one of the most usable and profitable plants for industrial applications. This family of plants has been cultivated and used for millennia, there existing around 230 species perfectly adapted to both cold and warm weather throughout all the continents [31]. It shows a worldwide production of approximately 3 million tons in 2017, being Canada, China, India, USA, and Ukraine the main producers [32]. Linen varieties are usually classified as flax, for bast fiber production in the textile industry, and linseed, for seed oil extraction in the chemical and food industry. Flax fiber (FF) stands out due to its high cellulose percentage (~78 wt%) [33] and the disorientation of its cellulose microfibrils, which show an angle of 10° with the longitudinal axis of the cell [34]. These properties translate into a high mechanical performance compared to other vegetables fibers and makes FF an interesting natural sourced option for fabrics. Indeed, FF is one of the oldest plant fibers used in household textiles or garments [35]. Furthermore, it can also be applied as a suitable reinforcement to replace glass fiber (GF) in polymer composites [36]. Nowadays, nonwovens based on FFs are finding new applications in panel boards, insulation or asbestos, and novel multilayer packaging structures due to the importance of the eco-design [37].

However, it is widely known that polymer composites based on lignocellulosic particles or fibers habitually present low mechanical and thermomechanical performances due to the absence or poor interfacial adhesion between the fillers and the matrix. This issue has been ascribed to the difference in hydrophobicity between the polymers and the plant derived fillers [38,39,40]. Therefore, to achieve adequate performance, it is essential to carry out a filler surface pretreatment [41] or use a coupling agent [42] to ensure an enhanced adhesion with the polymer matrix. Furthermore, the modification of the natural fillers can also influence some characteristics such as their hydrophobicity and porosity [43]. These methods are generally based on cleaning and/or activating the filler surface and they include both the use of chemicals and physical methods (e.g., ultraviolet (UV) radiation) [44]. Among the different fiber surface pretreatments based on chemical substances, alkalization, also called “mercerization”, is the most standardized one due to it reduces hydrophilicity and additionally increases the effectiveness of further chemical treatments [45,46]. In this regard, silanes and chemically modified silanes are some of the most used coupling agents employed in polymer-reinforced composites [47]. Additionally, the use of reactive compatibilizers represents a straightforward strategy to enhance the particle–matrix adhesion. These additives are generally based on polymers with low molecular weight (*M*_w_), oligomers, or even, more recently, functionalized vegetable oils with several reactive groups that are prone to form chemical bonds during melt processing [48,49]. In this sense, some commercially available copolymers and oligomers based on epoxy, maleic, or glycidyl methacrylate groups have successfully compatibilized biopolyesters and their green composites with lignocellulosic particles [22,50,51,52].

The present study evaluated and compared the effect of surface treatments and different reactive compatibilizers on the performance of green composites based on PLA and short FFs prepared by twin-screw extrusion (TSE) and injection molding. To this end, the lignocellulosic fillers were initially subjected to an alkali pretreatment and drying. Then, four different compatibilization routes were explored, namely silanization and reactive extrusion (REX) with a random copolymer based on glycidyl methacrylate (GMA), an oligomer containing multiple epoxy groups, and linseed oil multi-functionalized with maleic anhydride (MA). The mechanical, morphological, thermal, and thermomechanical properties of the injection-molded green composite pieces were analyzed and related to the different compatibilization strategies explored.

## 2. Materials and Methods

### 2.1. Materials

PLA Ingeo^TM^ Biopolymer 6201D was obtained from NatureWorks LLC (Minnetonka, MN, USA). It was supplied in pellets, having a melt flow rate (MFR) of 15–30 g/10 min measured at 210 °C and 2.16 kg and a true density of 1.24 g/cm^3^. This resin is suitable for injection molding and it has been previously screened for preparing green composites due to its high fluidity [19,23,53,54]. Schwarzwälder Textil-Werke Heinrich Kautzmann GmbH (Schenkenzell, Germany) delivered FF as NATURAL FIBRE. The manufacturer supplied the fibers with a length of 6 mm in yarns, having the aspect of a “wool ball”, as a processing by-product of the linen industry. A natural scale view of the as-received short FFs is shown in Figure 1.

Sodium hydroxide (NaOH), also known as caustic soda, with a purity of 99%, was obtained from Cofarcas, S.A. (Burgos, Spain) and it was used for preparing the alkali solution. For the silanization treatment, (3-glycidyloxypropyl) trimethoxysilane (GPTMS), also called GLYMO, with CAS 2530-83-8, was obtained from Sigma-Aldrich S.A. (Madrid, Spain). It shows a *M*_W_ of 236.34 g/mol and a density of 1.07 g/${\mathrm{cm}}^{3}$.

Joncryl^®^ 4368C, a multi-functional epoxy-based styrene-acrylic oligomer (ESAO), was supplied in solid flakes by BASF S.A. (Barcelona, Spain). This petrochemical oligomer presents values of *M*_w_ of 680 g/mol, *T*_g_ of 54 °C, and an epoxy equivalent weight (EEW) of 285 g/mol. The manufacturer recommends a dosage of 0.1–1 wt % for appropriate processability of biodegradable polyesters and complying with food contact regulations. Xibond^TM^ 920, a poly(styrene-*co*-glycidyl methacrylate) random copolymer (PS-*co*-GMA), was obtained from Polyscope (Geleen, The Netherlands). This petrochemical copolymer has a *M*_w_ value of 50,000 g/mol and a glass transition temperature (*T*_g_) of 95 °C. The manufacturer recommends a dosage level of 0.1–5 wt %, while not exceeding 330 °C to avoid thermal degradation. Maleinized linseed oil (MLO), CAS number 68309-51-3, was provided by Vandeputte (Mouscron, Belgium) as VEOMER LIN. It presents an acid value of 105–130 mg potassium hydroxide (KOH)/g and a viscosity of approximately 1000 cP at 20 °C.

### 2.2. Pretreatment of the Flaxseed Fibers

The as-received short FFs were first washed in distilled water to remove residuals and/or dirtiness. Then, the fibers were subjected to an alkali pretreatment in a water bath containing NaOH at 2 wt % at room temperature for 1 h, following the procedure described by Sever et al. [55]. Thereafter, the fibers were further washed with distilled water repeatedly and dried at 60 °C for 24 h in an air circulation oven Carbolite model PN from Carbolite Gero Ltd. (Hope Valley, UK).

Part of the alkali-pretreated FFs was also subjected to silanization. To this end, the alkalized fibers were dipped in distilled water bath containing 1 wt % of GPTMS, referred to the total amount of FF, and then magnetically stirred for 2 h at room temperature. After this, the silanized FFs were removed from the bath and dried in a stove at 50 °C for 2 days. The concentration of the glycidyl-silane reactive coupling agent was selected based on our previous studies [56,57,58].

### 2.3. Reactive Extrusion of the Green Composites

Prior to melt processing, the PLA pellets were dried in a dehumidifier MDEO from Industrial Marsé (Barcelona, Spain) at 60 °C for 24 h to eliminate residual moisture. Then, the PLA pellets and the pretreated FFs were mixed and fed at a constant ratio of 20 wt % into the main hopper of a twin-screw co-rotating extruder from Dupra S.L. (Alicante, Spain). The extruder features two screws of 25 mm diameter (D) with a length-to-diameter ratio (L/D) of 24. The modular barrel is equipped with 4 individual heating zones coupled to a strand die. Figure 2 shows the extruder layout with its different sections and screw configuration. In the feed section, the pellets enter the barrel with the rotating screws. The compression section contains two mixing zones, a conveying section where the pitch reduces in order to shear the molten plastic at the barrel wall followed by a series of kneading blocks (KBs, 8 × 90° R and 5 × 44° L) to enhance filler dispersion. In the metering zone, the channel depth is reduced and the melts moves in a spiraling motion down the screw channels. The melt is finally pumped out of the extruder through an annular die.

The three tested compatibilizers were added as different parts per hundred resin (phr) of composite. Particularly, PS-*co*-GMA and ESAO were both added at 1 phr, as this content has reported to enhance the interaction between biopolymers without leading undesired effects such as gel formation [50,59]. The selected MLO content was 5 phr since higher oil loadings could exert a negative effect by saturation [60]. A neat PLA sample was also prepared in the same conditions as the control material. In all cases, temperature profile was set at 165 °C at the feed section, 170 °C, 175 °C, and 180 °C at the die. The rotating speed was adjusted to 16 rpm and the residence time was kept at approximately 1 min, being estimated by means of a blue masterbatch [20]. The extruded strands were cooled down in air and granulated in pellets by an air-knife unit. Table 1 summarizes the pretreatments and compositions for all the compounded materials.

### 2.4. Injection Molding of the Green Composites

The resultant pellets were dried at 60 °C for 72 h to remove moisture since PLA has a high sensitivity to hydrolysis. Each sample was processed by injection molding in a Meteor 270/75 from Mateu & Solé (Barcelona, Spain). The temperature profile was set, from the feed section to the injection nozzle, as 180–175–170–170 °C. The cavity filling and packing times were 1 s and 10 s, respectively. The clamping force was 75 tons and the samples were cooled in the mold for 10 s. Pieces with a thickness of approximately 4 mm were obtained for characterization.

### 2.5. Characterization of the Green Composite Pieces

#### 2.5.1. Microscopy

Optical microscopy of the as-received FFs was performed using a stereomicroscope system SZX7 model from Olympus (Tokyo, Japan) with an ocular magnifying glass of 12.5× and equipped with a KL1500-LCD light source. The morphology of the as-received FFs and the fracture surfaces of the injection-molded PLA and PLA/FF pieces obtained from the impact strength test were also observed by field emission scanning electron microscopy (FESEM) in a ZEISS ULTRA 55 FESEM microscope from Oxfrod Instruments (Abingdon, UK). The microscope worked at an acceleration voltage of 2 kV. Prior to analysis, the fracture surfaces were coated with a gold-palladium alloy in a Quorun Technologies Ltd. EMITECH mod. SC7620 sputter coater (East Sussex, UK).

#### 2.5.2. Mechanical Tests

Tensile tests of the PLA and PLA/FF composite pieces were carried out in a universal test machine ELIB 30 from S.A.E. Ibertest (Madrid, Spain) using samples sizing 150 mm × 10 mm × 4 mm and following the guidelines of UNE-EN-ISO 527-1 standard. The selected load cell was 5 kN whereas the cross-head speed was set to 5 mm/min. Impact strength was tested in a 6-J Charpy pendulum from Metrotec S.A. (San Sebastian, Spain) using unnotched pieces with dimensions of 80 mm × 10 mm × 4 mm and as indicated in the ISO 179. Finally, hardness was measured in a Shore D durometer mod. 673-D from J. Bot S.A. (Barcelona, Spain), following the ISO 868 standard. At least six different samples were tested for each mechanical test.

#### 2.5.3. Thermal Tests

Thermal transition of the PLA and PLA/FF composites was studied by differential scanning calorimetry (DSC) in a Mettler-Toledo 821 calorimeter (Schwerzenbach, Switzerland). An average weight of 5–7 mg of each sample was placed in 40-µL aluminum-sealed crucibles and subjected to a heating ramp from 25 °C to 200 °C at 10 °C/min in inert atmosphere of nitrogen with a constant flow of 30 mL/min. The percentage of crystallinity ($X_{c}$) was calculated using Equation (1):(1)$$X_{C}=\left[\frac{\Delta H_{m}-\Delta H_{CC}}{\Delta H_{m}^{0}\cdot \left(1-w\right)}\right]\times 100$$
where ∆*H*_m_ (J/g) and ∆*H*_CC_ (J/g) correspond to the melting and cold crystallization enthalpies of PLA, respectively. ∆*H*_m_^0^ (J/g) is the theoretical value of a fully crystalline PLA, that is, 93.0 J/g [61] and 1 − *w* indicates the weight fraction of PLA in the green composites.

Thermogravimetric analysis (TGA) was carried out to evaluate the thermal stability of the PLA and PLA/FF composite pieces. A TGA/SDTA 851 thermobalance from Mettler-Toledo Inc. (Schwerzenbach, Switzerland) was employed using a single-step thermal program from 30 to 700 °C at a heating rate of 20 °C/min in an air atmosphere. Both thermal analyses were performed in triplicate.

#### 2.5.4. Thermomechanical Tests

The effect of temperature on the mechanical properties of the PLA and PLA/FF composite pieces was studied by dynamic mechanical thermal analysis (DMTA) in an oscillatory rheometer AR-G2 from TA Instruments (New Castle, DE, USA) equipped with an especial clamp system for solid samples that works with a combination of shear-torsion stresses. Pieces with a dimension of 40 mm × 10 mm × 4 mm were tested and the maximum shear deformation (%) was set to 0.1% at a frequency of 1 Hz. The thermal sweep was scheduled from 30 °C to 140 °C at a heating rate of 2 °C/min. The DMTA tests were carried out in triplicate.

## 3. Results and Discussion

### 3.1. Morphology of the Flaxseed Fibers

Figure 3 shows the morphology of the as-received yarn of FF and the surfaces of the FFs after the alkali and subsequent silane pretreatments. In the optical microscopy image included in Figure 3a, one can see that the as-received yarn of FF consisted of long individual fibers with a heterogeneous fiber diameter. FESEM analysis of the alkali-pretreated FFs prior and after silanization, respectively, shown in Figure 3b,c, indicated that fiber diameter varied in the 10–25 μm range and it was unaltered during silanization. Images taken at higher magnification, that is, 1000×, which are included in Figure 3d,e, revealed that the fiber surface was rough and fluted. This morphology is in agreement with previous studies, indicating that fibers derived from the flax plant are grouped in conglomerates of elementary fibers bonded by a smooth pectin and lignin phase to form bast technical fibers with a diameter of 50–100 μm [62]. Furthermore, FFs are long plant cells in which perpendicular dislocations, generally termed “kink bands”, occur naturally but they can also be formed by physical stresses [63,64]. These kink bands were easily visible in the FESEM micrographs, being distributed randomly along the fiber axis. These dislocations are known to result in a reduction of the fiber tensile properties, and they, furthermore, contribute to a non-linear straining behavior [29,65].

As commented above, these natural fibers are composed of cellulose chains strongly linked to lignin and hemicellulose. The alkali pretreatment of the fiber was performed to remove the cemented material, that is, lignin and hemicellulose, and modified the cellulose structure [66]. For instance, Valadez-Gonzales et al. [46] showed that henequén fibers with a diluted alkaline solution promoted the partial removal of hemicellulose, waxes, and lignin on the fiber surface. This process led to some changes in their morphology and chemical composition. This particular microfibrillar structure can be observed in the FESEM micrographs shown in Figure 3b,d in which the alkali-pretreated FFs were rugged due the decomposition of its constituent compounds. In particular, the alkalized FFs showed clean and smoother surfaces without impurities, which prove the effectiveness of the diluted NaOH solution to improve the overall fiber surface quality. After silanization, one can observe in Figure 3c,e that the fibers showed a similar morphology but they developed rougher surfaces due to the formation of nano-sized precipitates that correspond to the spherical silane molecules. This surface has been specifically ascribed to the presence of thin silane layers produced by the formation of a siloxane bonds (Si–O–Si) through reaction between the hydrolyzed alkoxy groups (R–O) in the silane and hydroxyl functional groups (–OH) available on the fibers surface [57]. As a result of this surface modification, the FFs contained suitable glycidyl functional groups to react with the biopolymer matrix of the green composite.

### 3.2. Mechanical Properties of the Green Composite Pieces

Table 2 gathers the results of the mechanical tests of the injection-molded pieces of the neat PLA and the PLA/FF composites. In relation to the neat PLA, one can observe that the piece showed mechanical properties of a hard but brittle material with values of elastic modulus (E) of 1194.2 MPa, tensile strength at yield (σ_y_) of 64.7 MPa, and elongation at break (ε_b_) of 8.1%. Its brittleness was also reflected in terms of impact strength, showing a value of 34.5 kJ/m^2^. As one expected, the addition of the alkali-pretreated flax fibers yielded an increment in the rigidity and brittleness, resulting in green composite samples with an average E and σ_y_ values of ~1750 MPa and ~39 MPa, respectively. Moreover, the incorporation of the lignocellulosic fillers induced a reduction in ductility, showing a ε_b_ value of 3.4% and impact strength of 5.8 kJ/m^2^. Additionally, hardness increased from 75.8, for the neat PLA piece, to 79.5, for the PLA/FF composite piece. This ductility impairment can be ascribed to the hydrophilic behavior of the lignocellulosic fibers that may result in a poor dispersion and incompatibility with the hydrophobic PLA matrix, leading to a poor stress transfer between the two components of the green composite [67,68,69]. Although FF was subjected to an alkaline pretreatment, this modification was mainly aimed to remove impurities and made the fiber surface cleaner and rougher. Similar pretreatments have resulted in a higher aspect ratio of the fiber, which is a positive aspect for fiber reinforcement, but the particle and matrix remained incompatible since no chemical modification was performed [5,11].

One can observe that the green composite pieces prepared with the alkalized FFs subjected to silanization, the here so-called PLA/FF + GPTMS, showed a slight reduction in elasticity but a remarkable increase in strength and ductility when compared to the PLA/FF pieces. In particular, the pieces showed a value of E of 1676.9 MPa, still higher than that of PLA, and σ_y_ and ε_b_ values of 57.9 MPa and 5.7%, respectively, representing an increment of 48% and 67% compared with the uncompatibilized PLA/FF piece, although both values were still lower than those of the neat PLA. These results suggest that the glycidyl silane modification successfully promoted the interfacial interaction between FF and PLA. Other research works ascribed the effectiveness of silanization to the formation of Si–C, C–H, and N–H stretching bonds between the hydrolyzed silane and the terminal –OH groups present in both cellulose and PLA [22,70,71,72]. Therefore, this chemical bonding can potentially play a key role in forming a link between the lignocellulosic fibers and PLA, with consequent enhancement of the mechanical properties. This effect was also reflected in the impact strength, which showed a notably increment to 21.9 kJ/m^2^ and an increase in hardness, reaching a value of 83.1.

Furthermore, the incorporation of the two petrochemical reactive additives, that is, PS-*co*-GMA and ESAO, also produced an improvement in the mechanical properties of the green composite. The addition of ESAO yielded pieces with E and σ_y_ values of 1685.1 MPa and 65.2 MPa, respectively, while ε_b_ significantly increased to 6.8%. Alternatively, the green composite pieces processed with PS-*co*-GMA showed E and σ_y_ values of nearly 1720 MPa and 62 MPa, respectively, while ε_b_ was 5.8%. Toughness also improved in both formulations, showing values of 24.1 kJ/m^2^, for the pieces processed with ESAO, and 21.3 kJ/m^2^, for those with PS-*co*-GMA. The hardness values were also higher in the case of ESAO, showing values of approximately 83.2 and 78.8, respectively. Therefore, although both reactive additives enhanced the mechanical performance of the green composites, the improvement achieved by the addition of ESAO was slightly higher than that attained with PS-*co*-GMA. This difference can be ascribed to both the different chemical reactive groups and also the differences in the *M*_w_ of these two additives [68,73]. In the case of PS-*co*-GMA, Anakabe et al. [74] reported that the –OH end groups of PLA can react with the MA groups present in the random copolymer, leading to ester and ether linkages. For ESAO, its multiple epoxy groups can react with the carboxyl (–COOH) and –OH end groups of the PLA chains based on a linear chain-extended, branched or even cross-linked structure depending on the functionality whereas the remaining epoxy groups are also able to react with the –OH groups present on the surface of the cellulosic fillers [73,75]. In this regard, Battegazzore et al. [76,77] reported the ability of ESAO to increase the interfacial adhesion between cotton fabric with PLA/poly(3-hydroxybutyrate-*co*-3-hydroxyhexanoate) (PHBH) blends as well as to produce multilayer cotton fabric composites of PLA/PHBH with less porosity and improved stress transfer capacity. This dual reactivity of ESAO of coupling agent and melt strength additive can potentially lead to green composites of higher improvement. Furthermore, the lower *M*_w_ of oligomer could also favor its higher miscibility and reactivity within the PLA matrix.

In relation to the oil plant derived multi-functionalized oil, the green composites pieces processed with MLO presented significantly lower values of E and σ_y_, that is, 1192.7 MPa and 24.8 MPa, respectively. However, interestingly, the ε_b_ value raised to 10.7%, being the highest value attained for the green composite in the present study. This ductility enhancement can be ascribed to the plasticization of the PLA matrix in which the molecular forces are reduced and the free volume increased [78]. Indeed, MLO has been reported as an effective plasticizer for PLA materials, promoting an additional interfacial adhesion enhancement with both organic and inorganic fillers based on its multi-functionality [22,56,60]. In particular, the MAH groups present in MLO can result in the formation of carboxylic ester bonds by their reaction with the –OH end groups of both the PLA molecules and cellulose, resulting in a cellulose-grafted PLA (cellulose-*g*-PLA) structure [22]. It is also worthy to note that the green composite pieces processed with MLO showed the lowest value of impact strength, that is, 20.4 kJ/m^2^, and also hardness, that is, 76.2, among all the tested compatibilized green composites. This may seem contradictory in relation to values of ductility, but other mechanical properties such as tensile strength also influences in toughness.

### 3.3. Morphology of the Green Composite Pieces

The FESEM images taken at 1000× of the fracture surfaces after the impact test of the pieces are shown in Figure 4. The micrograph shown in Figure 4a corresponds to the PLA sample, in which a smooth and homogeneous surface with small plastic deformation filaments can be observed. This morphology correlates well the previous mechanical tests that indicated that the PLA piece was brittle. Figure 4b shows the fracture surface of the PLA/FF piece. One can observe that, even though the fibers were pretreated with NaOH, their interaction with the PLA matrix was low. A considerable gap between the lignocellulosic fibers and the biopolyester matrix can be noticed as well as some holes due to fiber detachment during fracture. This type of morphology has been previously described for green composites, reflecting an increase in rigidity due to fiber reinforcement but also reduction in ductility since the stress is not effectively transferred from the matrix to the fillers [70,79].

As can be seen in Figure 4c, the silanization performed on FFs promoted a high interfacial interaction with the PLA matrix due to the nearly absence of gap. A similar interfacial enhancement has been achieved for kenaf, pineapple, and *Phormium Tenax* (New Zealand flax) fibers subjected to alkaline and silane combined pretreatments [70,71,72,79]. The FESEM micrograph shows that some fibers remained embedded in the PLA matrix after fracture, which further suggests their strong adhesion to the matrix. This morphology can be correlated with the mechanical improvement described above, in which the green composite presented high strength and toughness. However, the fracture surface was relatively smooth, similar to that of the neat PLA piece, indicating that the fiber-biopolymer interface was not strong enough to modify the fracture energy. It has been reported that covalent bonds between the fillers and matrix created by special reactive surface treatments are very strong (60–80 kJ/mol), but in practice the strength of the interaction is somewhere between this and secondary and weaker van der Waals forces such as dipole–dipole (Keesom), induced dipole (Deby), and dispersion (London) interactions so that the ultimate mechanical strength of the composite is lower and more complex than the one expected [80]. Figure 4d shows that the addition of ESAO produced a morphological change in the fracture surface of the green composite piece. In particular, the incorporation of the reactive oligomer yielded a rough and irregular surface fracture. In addition, the stress cracks were shorter, indicating that the green composite developed a fracture with certain plastic deformation. However, the FESEM micrograph revealed that the interface between the PLA matrix and the lignocellulosic fibers was not promoted to a high extent and the gap around the embedded fibers was higher than that observed in the PLA/FF + GPTMS piece. Moreover, the presence of large cylindrical cavities indicates that fibers detached from the matrix during fracture. Therefore, the mechanical improvement attained in these sample pieces can be more related to the effect of ESAO as chain extender than an interfacial adhesion improvement, since it can contribute to an increase in the mechanical strength and favor a crack arresting mechanism by an increase of the PLA’s *M*_w_ [81]. In relation to PS-*co*-GMA, the fracture surface of the green composite piece processed with the random copolymer is displayed in Figure 4e. The morphology of the PLA/FF + PS-*co*-GMA piece was rough, being relatively similar to that of PLA/FF + ESAO; however, the crack edges appeared more pronounced. This small difference in the green composite morphology correlates well with the mechanical results, showing that the green composite pieces processed with PS-*co*-GMA were slightly more rigid but also less tough. Finally, in Figure 4f, the fracture surface of the green composite piece processed with MLO is shown. One can observe that the multi-functionalized vegetable oil caused a remarkable change in the fracture surface morphology. In particular, the MLO incorporation promoted the formation of a smooth surface with considerable number of small cavities homogeneously distributed along the PLA matrix, causing a porous-like surface. Similar morphologies were reported in previous studies for MLO-containing PLA materials [22,60]. Other authors described a similar phenomenon for plasticizers due to they form a separated spherical plasticizer-rich phase that result in the creation of holes during fracture [82,83,84]. One can also observe that the gaps existing around fibers were shorter compared with those observed in the uncompatibilized PLA/FF piece, but larger than those attained in the other compatibilized composite pieces. This indicates that MLO mainly induced a matrix modification and it did not effectively enhance the interfacial adhesion as much as the other compatibilizers, which is in agreement with the mechanical analysis described above.

### 3.4. Thermal Properties of the Green Composite Pieces

Figure 5 shows the DSC thermograms during the first heating of the neat PLA piece and the PLA/FF pieces. The average values of the main thermal transitions obtained from the DSC curves are gathered in Table 3. The enthalpies associated with the cold crystallization and the melting processes are also reported in the table. A step change in the base lines in the 60–70 °C range can be observed, which corresponds to the *T*_g_ of PLA. For the neat PLA sample, the mean *T*_g_ value was located at 67.3 °C. Then, the exothermic peaks located between 110 °C and 130 °C can be attributed to the cold crystallization temperature (*T*_cc_) of PLA, which was 114.7 °C for the neat piece of PLA. Finally, at higher temperatures, in the range of 150–180 °C, the endothermic peaks represent the melting process of the crystalline PLA domains. In the thermogram of the neat PLA sample, one can observe the presence of two overlapped peaks. The first one appeared at 167.7 °C and the second one at 174.1 °C. This double-peak phenomenon is ascribed to crystal reorganization upon melting in polyesters, by which imperfect crystals melt and the amorphous regions are ordered into spherulites with thicker lamellar thicknesses that thereafter melt at higher temperatures [59,85].

As can be observed in the thermograms, the incorporation of the alkalized FFs produced a slight reduction of the *T*_g_ value, which remained at 67 °C. The cold crystallization process, however, occurred at a lower temperature, that is, 109 °C. This reduction of the *T*_cc_ value indicates that FF could act as external nuclei for the formation of PLA crystallites. Indeed, the nucleating effect of lignocellulosic particles on PLA has been widely reported [86,87]. It is also worth noting that the presence of FFs also slightly reduced the percentage of crystallinity, which suggests that the crystallites formed were less perfect due to the presence of the fillers. In this regard, Huang et al. [88] reported a similar phenomenon as a result of the interfacial interaction of the fillers with the matrix that could favor crystal nucleation but reduce crystal growth. This phenomenon was further correlated with the observed reduction in the *T*_m_ values, that is, 154.3 and 164.9 °C, since less perfect crystals underwent the melting process. One should also mention that double melting continued to be observable, confirming the absence or low interaction between the fillers and the PLA matrix.

In all the compatibilized PLA/FF composites, the *T*_g_ values remained in the 64–65 °C range, with the exception of MLO. In the case of the green composite piece processed with the multi-functionalized vegetable oil, *T*_g_ decreased to 61.7 °C. This observation further supports that MLO plasticized the PLA matrix by reducing the interaction between the biopolymer chains and increasing their mobility [84]. This phenomenon was well evidenced by Ferri et al. [60], who showed that the reduction of the *T*_g_ values fully correlated with the amount of MLO added. With regard to the cold crystallization process, an increase in the *T*_cc_ values was observed for all the compatibilized formulations when compared with the uncompatibilized green composite. This result points out that the compatibilizers could partially occupy the free volume between FFs and the PLA molecules and restricted the nucleation of the fillers. Furthermore, in the case of the green composite samples processed with ESAO, PS-*co*-GMA and, even, MLO, this crystallinity restriction can be related to the formation of chain-extended and/or branched structures of PLA with a higher impairment to crystallize [74,89]. Some authors have additionally concluded that this reaction can create cross-linked structures that could lead to difficulties in the chains mobility and, hence, cold crystallization is delayed [50,74,90]. In the case of the PLA/FF + MLO piece, showing the highest *T*_cc_ value, that is, 124.5 °C, the newly formed spherical-shape phases dispersed in the PLA matrix could further contribute to delay crystallization of PLA. One can also observe that all the compatibilizers suppressed the double-melting peak phenomenon and PLA melted in a single peak in the 151–154 °C range. This result suggests that all the crystalline structures presented similar lamellae thicknesses, being lower than that of PLA in the pieces without compatibilizer due to the effect of those on the PLA chains [53]. The degree of crystallinity of PLA in the green composite pieces slightly increased when the fibers were silanized or processed with the reactive compatibilizers. This effect has been related to a higher fiber-to-matrix adhesion but it could also be due to an improvement of *M*_w_ by a chain extension and/or prevention of random chain scission reactions (e.g., hydrolysis) by which more mass of biopolymer crystallized in the mold [59].

Figure 6 shows the TGA curves of the FF, the neat PLA piece, and the green composite pieces uncompatibilized and compatibilized by the different methods. The thermal stability values extracted from the TGA curves are summarized in Table 4, which gathers the temperature required for a loss of weight of 5% (*T*_5%_) that is representative for the onset of degradation, the degradation temperatures (*T*_deg1_ and *T*_deg2_), and the amount of residual mass at 700 °C. Figure 6a shows the mass loss as a function of temperature, whereas, in Figure 6b, their respective DTG curves are included.

One can see that thermal degradation of the alkali-pretreated FF took place in three main steps. Initially, at a temperature close to 100 °C, a slight mass loss of ~3.5% was observed, which corresponds with the moisture and remaining solvent retained in the fibers. Following the TGA curve, a change in the slope of the curve was observed when temperature reached ~255 °C. This represents a mass loss of ~60% and it finalized at ~390 °C. Previous studies reporting the thermal decomposition of lignocellulosic particles have attributed this mass loss to the oxidation of their main constituents, primarily in the amorphous regions (hemicellulose), followed by cellulose and lignin. This phenomenon is normally referred as the “active pyrolysis zone” and it is followed by a second degradation step, denominated “passive pyrolysis zone”, where the oxidation of the residual char occurs [91]. In the present FF curve, this pyrolysis step was observable in the 400–600 °C range. The TGA curve of FF is totally in agreement with others reports focused on lignocellulosic fibers [92]. For temperatures above 600 °C, thermal degradation continued but no significant weight losses were detected, indicating that the entire organic component were already pyrolyzed. The residual mass at the end of the analysis was over 20%, which can be related to the high mineral content present in flax [53].

Neat PLA showed a typical thermal degradation for a biopolyester based on two steps. In the range from nearly 335 °C to 390 °C, around 95% of the mass was lost. This peak was sharp and intense due to the chain-scission reaction through breakage of its ester groups [93]. After this, a second low-intense degradation stage took place, which was more detectable in the DTG graph. This corresponds with the fully oxidation of the char residue produced by the formation of some complexes, C = C bonds, aromatic species, etc. during the first stage [94]. After the incorporation of the alkalized FFs, the thermal stability of PLA was reduced. Both main thermal stability values, that is, *T*_5%_ and *T*_deg1_, were reduced by approximately 55 °C and 33 °C, respectively. The lower thermal stability of green composites has mainly been explained in terms of the low thermal stability of the lignocellulosic fillers and also to the remaining moisture, which is extremely difficult to remove [53]. Moreover, FF increased the intensity of the second degradation peak centered at ~410 °C, which can be related to the passive pyrolysis of the lignocellulosic fillers. In relation to the compatibilized PLA/FF samples, all coupling strategies yielded a similar positive effect on the thermal stability of the green composite. In particular, the *T*_5%_ values ranged from 330 °C to 340 °C, while *T*_deg1_ was in the 375–385 °C range. Therefore, the use of different compatibilizers resulted in green composites with thermal stability values in the same range or even higher than those observed for neat PLA. This thermal enhancement can be related to the improved interfacial adhesion achieved in the green composite due to the chemical interaction established between the lignocellulosic fillers and the biopolymer matrix. As in the mechanical properties, the highest improvement was observed for the green composite processed with ESAO, although the thermal stability of all the compatibilized samples was relatively similar and close to that of neat PLA.

### 3.5. Thermomechanical Properties of the Green Composite Pieces

As shown in Figure 7, the thermomechanical behavior of the PLA and PLA/FF pieces was studied by the variation of the storage modulus (G′) and the damping factor (tanδ) as a function of temperature. Figure 7a gathers the evolution of G′ with temperature. For all the PLA pieces, it can be observed a relatively high stiffness at low temperatures and then a sharply decrease of the G′ values between 50 °C and 80 °C due to the temperature reached the alpha (α)-relaxation of the biopolymer. With the temperature increment, in the 85–100 °C range, the cold crystallization phenomenon occurred, as previously discussed during the DSC analysis. Table 5 shows the G′ values of the storage modulus measured at 40 °C, 75 °C, and 110 °C, being these temperatures representative of the stored elastic energy of the material in the three mentioned states, that is, amorphous glassy, amorphous rubber, and semi-crystalline. One can observe that all the green composite pieces presented higher values of G′ than the neat PLA piece. In particular, at 40 °C, the G′ of the PLA/FF piece was 1080.3 MPa, which represents almost the double of the unfilled PLA, that is, 566.5 MPa. This is consistent with the above-reported effect of mechanical reinforcement of PLA by the lignocellulosic fibers. Similar remarkable increases in stiffness for natural filler-reinforced polymer composites have been reported previously [22,48,56]. The reinforcing effect was more pronounced after the glass transition region, in which the G′ value measured at 75 °C increased from 2.2. MPa, in the neat PLA piece, to 13.1 MPa, in the PLA/FF piece. This points out the high effect of relative hard fillers in a rubber-like matrix. The G′ value also showed a nearly three-fold increased, from 47.7 MPa to 154.6 MPa, at 110 °C, after cold crystallization, when the alkali-pretreated FF were incorporated into PLA. One can also notice in the DMTA curves that, as also observed during the DSC analysis, the fillers presence also promoted the occurrence of the cold crystallization phenomenon at lower temperatures.

The use of the different compatibilizers also yielded higher G′ values at low temperatures, with the exception of MLO due to its secondary role as plasticizer. This observation confirms that all the here-tested compatibilization strategies successfully increased the adhesion of the alkali-pretreated FFs to the PLA matrix in the green composites. The highest G′ values were attained for the green composite pieces processed with the alkalized FFs that were further pretreated with GPTMS. In particular, their G′ value was nearly 3 GPa at 40 °C, being around three times higher than that of the PLA/FF pieces and representing an increment of ~500% compared with the neat PLA pieces. This trend was also kept at higher temperatures, showing G′ values of 26.7 MPa and 293.8 MPa at 75 °C and 110 °C, respectively. Some authors have concluded that silanization of lignocellulosic fibers can induce a noticeable increment in the mechanical rigidity of the polymer composites, though it also reduces the elastic limit, especially when the fibers are subjected to alkaline pretreatment [70,79]. This combination of surface pretreatments can notably modify and/or remove some substances present on the fiber surfaces (e.g., hemicellulose, cellulose, and lignin), which can favor the interaction with the polymer molecules, but it can also cause damage in the fiber due to an excess of delignification [95]. This statement is in agreement with the FESEM images shown above, in which one can observe that the fibers were rugged due the decomposition of its constituent compounds and rougher by the presence of the silane layer. Therefore, the structural modification of the fibers played a key role in the thermomechanical properties of the green composites This explanation further supports both the morphological properties, which showed that the fibers were highly adhered to the biopolyester matrix, but also the fact that the mechanical strength and toughness was lower when compared with ESAO and PS-*co*-GMA since in the latter cases the fiber morphology was better preserved. The slight difference of the pieces in terms of their mechanical and thermomechanical performance can also be related to the type of experiment since in the tensile test the effort was uniaxial, while in the DMTA tests the pieces were subjected to a combination of shear-torsion stresses and the contribution of the fiber-to-adhesion strength was higher. It is also worth noting that both reactive compatibilizers, in particular PS-*co*-GMA, delayed cold crystallization of PLA due to the above-described restriction in chain mobility caused by the formation of a macromolecular structure with higher *M*_w_.

Figure 7b represents the evolution of tanδ versus temperature. The peaks of the graphs represent the change in the thermomechanical behavior when the α-relaxation of the biopolymer is reached, which relates to its *T*_g_. As shown in Table 5, the peak for the neat PLA piece was located at 68.4 °C, which is similar to the *T*_g_ value reported by DSC. One can also observe that the green composite pieces showed lower peak values for α-relaxation, centered at ~63 °C, being also very similar to the values of *T*_g_ attained by DSC. In the case of the green composite processed with MLO, this value was 60.8 °C due to the plasticizing effect of the multi-functionalized vegetable oil. It should also be noted that all green composite samples had α-relaxation peaks with lower intensities than the neat PLA sample. This observation points out that the relaxation of the PLA chains was partially suppressed by the presence of the alkalized FFs and, hence, lower number of PLA chains underwent glass transition [44]. This fact also correlates well the crystallinity results obtained by DSC that showed that PLA developed slightly higher crystallinity in the compatibilized green composite pieces.

## 4. Conclusions

Four different routes were tested to improve the overall performance of injection-molded pieces of green composites made of PLA and short FFs. These strategies included a silanization pretreatment of alkalized FFs with GPTMS and the use of three reactive compatibilizers added during melt extrusion, that is, ESAO, PS-*co*-GMA, and MLO. The morphological analysis showed that the highest interfacial adhesion was obtained in the green composite pieces containing the fibers pretreated with GPTMS due to the glycidyl-silane reactive coupling agent was able to react with the –OH terminal groups present in both cellulose and PLA. Although the mechanical and thermal performance of the green composite pieces was improved, the fiber–biopolymer interface was not strong enough to modify the fracture energy and the improvement was relatively low. This effect was related to the fiber morphology damage that occurred due to an excess of delignification during silanization. In relation to the two petroleum derived reactive additives, that is, ESAO and PS-*co*-GMA, it was observed that these led to the highest mechanical resistance and toughness improvement and also the highest thermal stability, showing the multi-functional oligomer the most optimal properties. Since the morphological analysis revealed that the filler–matrix was relatively low, the enhancement attained was mainly ascribed to a matrix modification by the formation of a chain-extended, branched or even cross-linked structure through the functional groups present in both additives that can react with –OH groups of PLA. Regarding the multi-functionalized vegetable oil, the green composite processed with MLO also showed improved mechanical and thermal properties, showing the highest ductility. However, it was also observed that it plasticized the PLA matrix and, as a result, the mechanical and thermomechanical properties were not as high as those attained by means of the other strategies. Despite their relatively low properties, the MLO-containing green composites are still great candidates for more flexible applications and it is also the only solution fully based on natural raw materials. According to the here-attained results, one can conclude that the compatibilizer of highest potential for green composites is ESAO since it allows to produce, at very low contents, high-performing bio-based and biodegradable materials in terms of mechanical resistance, toughness, and thermal stability.

## Figures and Tables

**Figure 1 polymers-12-00821-f001:**
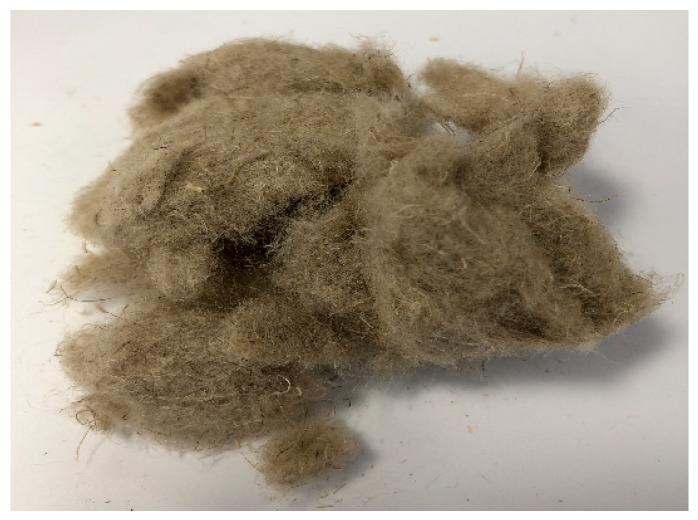
As-received yarn of short flaxseed fibers (FFs).

**Figure 2 polymers-12-00821-f002:**
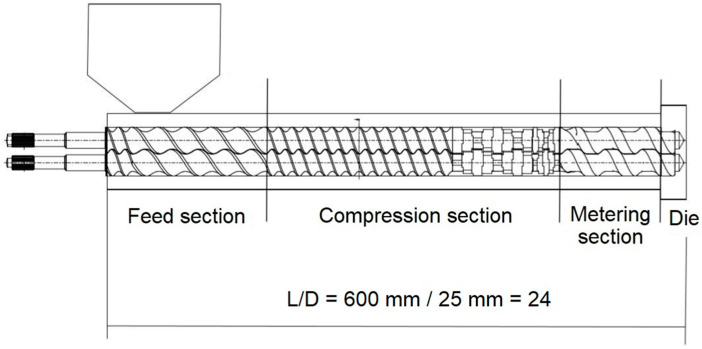
Twin-screw co-rotating extruder and its screw configuration.

**Figure 3 polymers-12-00821-f003:**
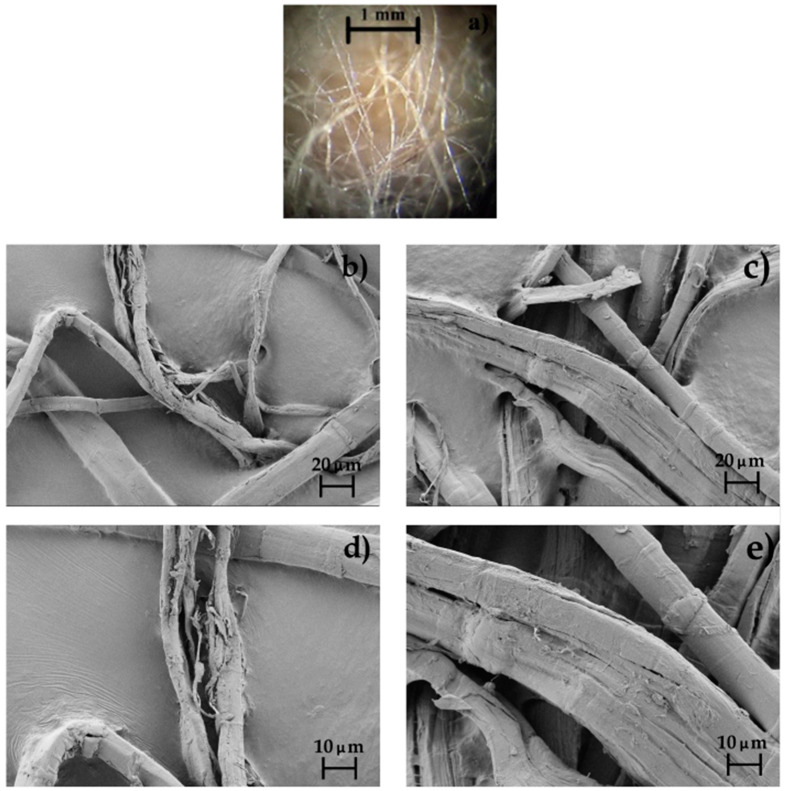
Morphology of the: (**a**) As-received yarn of flaxseed fibers (FFs) taken by optical microscopy at 12.5× with scale maker of 1 mm; (**b**) Alkali-pretreated FFs; (**c**) Alkali-pretreated FFs after silanization taken by field emission scanning electron microscopy (FESEM) at 500× with scale makers of 20 μm; (**d**) Alkali-pretreated FFs; (**e**) Alkali-pretreated FFs after silanization taken by FESEM at 1000× with scale makers of 10 μm.

**Figure 4 polymers-12-00821-f004:**
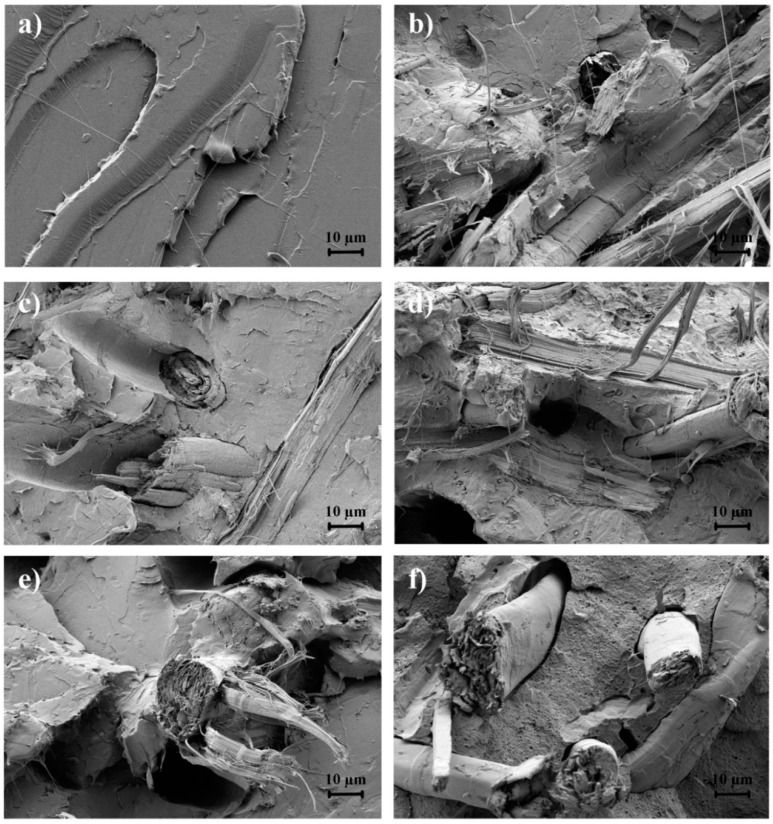
Field emission scanning electron microscopy (FESEM) images of the fracture surfaces of the polylactide (PLA)/flaxseed fiber (FF) pieces of: (**a**) PLA; (**b**) PLA/FF; (**c**) PLA/FF + (3-glycidyloxypropyl) trimethoxysilane (GPTMS); (**d**) PLA/FF + epoxy-based styrene-acrylic oligomer (ESAO); (**e**) PLA/FF + poly(styrene-*co*-glycidyl methacrylate) (PS-*co*-GMA); (**f**) PLA/FF + maleinized linseed oil (MLO). Images were taken at 1000× with scale makers of 10 µm.

**Figure 5 polymers-12-00821-f005:**
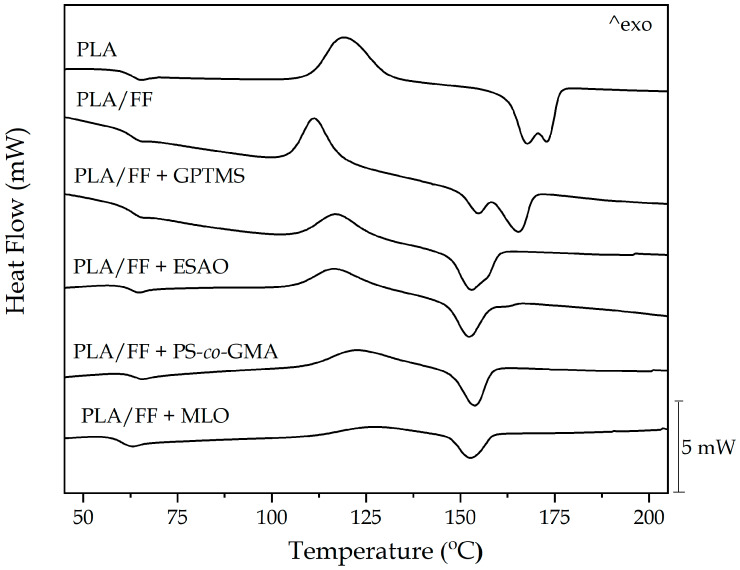
Differential scanning calorimetry (DSC) thermograms corresponding to the polylactide (PLA)/flaxseed fiber (FF) pieces compatibilized with (3-glycidyloxypropyl) trimethoxysilane (GPTMS), epoxy-based styrene-acrylic oligomer (ESAO), poly(styrene-*co*-glycidyl methacrylate) (PS-*co*-GMA), and maleinized linseed oil (MLO).

**Figure 6 polymers-12-00821-f006:**
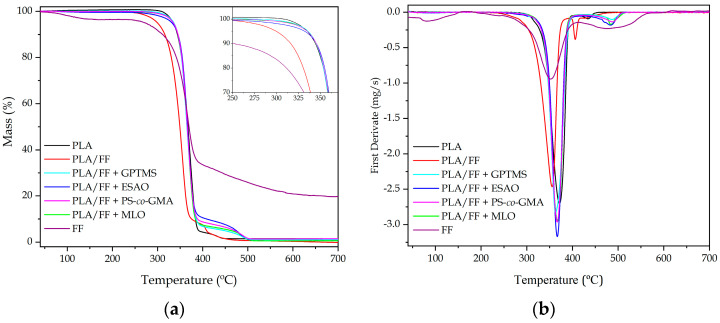
(**a**) Thermogravimetric analysis (TGA) curves with inset zooming the onset of degradation; and (**b**) first derivate thermogravimetric (DTG) curves corresponding to the polylactide (PLA)/flaxseed fiber (FF) pieces compatibilized with (3-glycidyloxypropyl) trimethoxysilane (GPTMS), epoxy-based styrene-acrylic oligomer (ESAO), poly(styrene-*co*-glycidyl methacrylate) (PS-*co*-GMA), and maleinized linseed oil (MLO).

**Figure 7 polymers-12-00821-f007:**
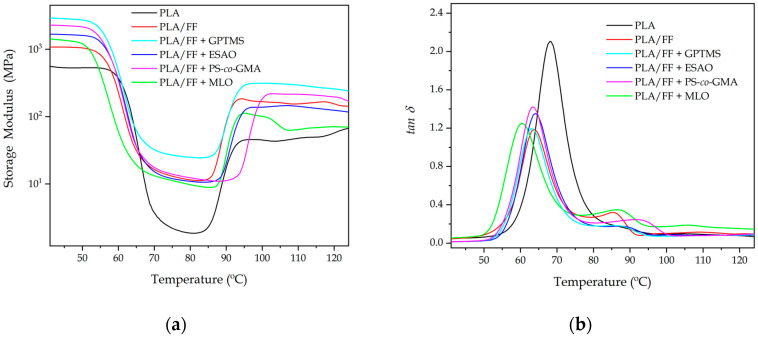
Evolution as a function of temperature of the (**a**) storage modulus and (**b**) dynamic damping factor (*tan δ*) of the polylactide (PLA)/flaxseed fiber (FF) pieces compatibilized with (3-glycidyloxypropyl) trimethoxysilane (GPTMS), epoxy-based styrene-acrylic oligomer (ESAO), poly(styrene-*co*-glycidyl methacrylate) (PS-*co*-GMA), and maleinized linseed oil (MLO).

**Table 1 polymers-12-00821-t001:** Code and composition of each sample according to the weight content (wt%) of polylactide (PLA) and flaxseed fiber (FF) in which epoxy-based styrene-acrylic oligomer (ESAO), poly(styrene-*co*-glycidyl methacrylate) (PS-*co*-GMA), and maleinized linseed oil (MLO) were added as parts per hundred resin (phr) of composite. Fiber pretreatments consisted of alkalization, in all cases, and silanization with (3-glycidyloxypropyl) trimethoxysilane (GPTMS).

Sample	PLA (wt %)	FF (wt %)	Fiber Pretreatment	Compatibilizer (phr)
PLA	100	0	-	-
PLA/FF	80	20	Alkalization	-
PLA/FF + GPTMS	80	20	Alkalization + Silanization	-
PLA/FF + ESAO	80	20	Alkalization	1
PLA/FF + PS-*co*-GMA	80	20	Alkalization	1
PLA/FF + MLO	80	20	Alkalization	5

**Table 2 polymers-12-00821-t002:** Mechanical properties of the polylactide (PLA)/flaxseed fiber (FF) pieces compatibilized with (3-glycidyloxypropyl) trimethoxysilane (GPTMS), epoxy-based styrene-acrylic oligomer (ESAO), poly(styrene-co-glycidyl methacrylate) (PS-*co*-GMA), and maleinized linseed oil (MLO) in term of elastic modulus (E), tensile strength at yield (σ_y_), elongation at break (ε_b_), impact strength, and Shore D hardness.

Piece	Tensile Test			Impact Strength (kJ/m^2^)	Shore D Hardness
	E (MPa)	σ_y_ (MPa)	ε_b_ (%)		
PLA	1194.2 ± 27.4	64.7 ± 1.2	8.1 ± 0.5	34.5 ± 2.7	75.8 ± 0.9
PLA/FF	1749.9 ± 32.9	39.1 ± 5.8	3.4 ± 0.2	5.8 ± 0.4	79.5 ± 1.4
PLA/FF + GPTMS	1676.9 ± 45.7	57.9 ± 4.9	5.7 ± 0.9	21.9 ± 2.1	83.1 ± 0.7
PLA/FF + ESAO	1685.1 ± 27.6	65.2 ± 4.1	6.8 ± 1.1	24.1 ± 2.7	83.2 ± 0.7
PLA/FF + PS-*co*-GMA	1719.7 ± 33.8	61.8 ± 5.2	5.8 ± 0.8	21.3 ± 3.7	78.8 ± 0.8
PLA/FF + MLO	1192.7 ± 41.4	24.8 ± 5.5	10.7 ± 1.1	20.4 ± 3.8	76.2 ± 0.9

**Table 3 polymers-12-00821-t003:** Thermal properties of the polylactide (PLA)/flaxseed fiber (FF) pieces compatibilized with (3-glycidyloxypropyl) trimethoxysilane (GPTMS), epoxy-based styrene-acrylic oligomer (ESAO), poly(styrene-*co*-glycidyl methacrylate) (PS-*co*-GMA), and maleinized linseed oil (MLO) in term of: glass transition temperature ($T_{g}$), cold crystallization temperature ($T_{\mathrm{cc}}$), melting temperature ($T_{m}$), cold crystallization enthalpy ($\Delta H_{\mathrm{cc}}$), melting enthalpy ($\Delta H_{m}$), and degree of crystallinity ($X_{c}$).

Piece	$T_{g}$ (°C)	$T_{\mathrm{cc}}$ (°C)	$T_{m}$ (°C)	$\Delta H_{\mathrm{cc}}\;(J/g)$	$\Delta H_{m}\;(J/g)$	$X_{c}\;(\%)$
PLA	67.3 ± 0.1	114.7 ± 0.5	167.7 ± 0.6/174.1 ± 0.4	28.61 ± 0.2	33.13 ± 0.2	4.86 ± 0.4
PLA/FF	67.0 ± 0.3	109.0 ± 0.4	154.3 ± 0.5/ 164.9 ± 0.4	19.50 ± 0.4	22.47 ± 0.2	3.99 ± 0.2
PLA/FF + GPTMS	64.7 ± 0.4	110.3 ± 0.4	151.3 ± 0.6	15.55 ± 0.5	19.88 ± 0.4	5.81 ± 0.5
PLA/FF + ESAO	64.0 ± 0.4	111.7 ± 0.6	151.0 ± 0.7	13.88 ± 0.6	17.48 ± 0.6	4.89 ± 0.3
PLA/FF + PS-*co*-GMA	64.3 ± 0.6	118.3 ± 0.2	153.2 ± 0.4	13.47 ± 0.4	18.32 ± 0.5	6.58 ± 0.4
PLA/FF + MLO	61.7 ± 0.2	124.5 ± 0.4	152.3 ± 0.4	10.48 ± 0.6	13.97 ± 0.4	4.92 ± 0.4

**Table 4 polymers-12-00821-t004:** Main thermal degradation parameters of the polylactide (PLA)/flaxseed fiber (FF) pieces compatibilized with (3-glycidyloxypropyl) trimethoxysilane (GPTMS), epoxy-based styrene-acrylic oligomer (ESAO), poly(styrene-*co*-glycidyl methacrylate) (PS-*co*-GMA), and maleinized linseed oil (MLO) in term of: temperature required for a loss of weight of 5% (*T*_5%_), degradation temperatures (*T*_deg1_ and *T*_deg2_), and residual mass at 700 °C.

Piece	*T*_5%_ (°C)	*T*_deg1_ (°C)	*T*_deg2_ (°C)	Residual Mass (%)
FF	255.3 ± 0.4	316.6 ± 0.4	419.8 ± 0.9	20.6 ± 0.2
PLA	334.2 ± 0.7	373.3 ± 0.4	410.3 ± 1.1	1.5 ± 0.3
PLA/FF	279.1 ± 0.6	340.6 ± 0.7	368.4 ± 1.1	3.4 ± 0.3
PLA/FF + GPTMS	337.7 ± 0.4	375.8 ± 1.1	411.6 ± 0.8	4.2 ± 0.4
PLA/FF + ESAO	332.2 ± 0.3	381.2 ± 0.9	412.7 ± 0.9	4.1 ± 0.3
PLA/FF + PS-*co*-GMA	331.6 ± 0.9	383.9 ± 1.2	411.2 ± 1.3	4.4 ± 0.2
PLA/FF + MLO	337.6 ± 0.5	381.7 ± 0.8	409.6 ± 1.2	3.6 ± 0.5

**Table 5 polymers-12-00821-t005:** Thermomechanical properties of the polylactide (PLA)/flaxseed fiber (FF) pieces compatibilized with (3-glycidyloxypropyl) trimethoxysilane (GPTMS), epoxy-based styrene-acrylic oligomer (ESAO), poly(styrene-*co*-glycidyl methacrylate) (PS-*co*-GMA), and maleinized linseed oil (MLO) in term of: storage modulus (G′) measured at 40 °C, 75 °C, and 110 °C and glass transition temperature (*T*_g_).

Piece	T_g_ (°C)	G′ (MPa)		
		40 °C	75 °C	110 °C
PLA	68.4 ± 1.3	566.5 ± 8.2	2.2 ± 0.7	47.7 ± 0.5
PLA/FF	63.8 ± 0.8	1080.3 ± 9.5	13.1 ± 0.5	154.6 ± 0.5
PLA/FF + GPTMS	63.1 ± 1.1	2908.8 ± 7.8	26.7 ± 0.7	293.8 ± 0.7
PLA/FF + ESAO	64.2 ± 1.2	1683.7 ± 7.3	11.9 ± 0.8	141.7 ± 0.5
PLA/FF + PS-*co*-GMA	63.4 ± 0.6	2293.4 ± 6.4	13.9 ± 0.8	214.2 ± 0.6
PLA/FF + MLO	60.8 ± 0.9	1437.6 ± 9.1	11.2 ± 0.7	63.6 ± 0.6

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
