# Peer review of "Evaluation of Different Compatibilization Strategies to Improve the Performance of Injection-Molded Green Composite Pieces Made of Polylactide Reinforced with Short Flaxseed Fibers"

_polymers, 2020, doi:10.3390/polym12040821_

Round 1

Reviewer 1 Report

In this manuscript, the performance of injection-molded pieces of polylactide reinforced with fibers is improved by the evaluation of different compatibilization strategies. Some interesting results are presented. However, moderate revision is needed to further improve the manuscript. For instance,
The abstract and conclusion should be improved and more concise, and the scientific principles should be further concluded in this work.
The authors should summarize the related attempts on the development of composite materials containing natural fillers with improved performance. It would be necessary to discuss by referring some articles, including: Compos Part A-Appl S, 2019, 124:105472.; Environ Sci Technol, 2019, 53: 6989-6996; J Manuf Process, 2019,45: 520-531.; Appl Clay Sci,2016, 132-133: 175-181.; Results Phys, 2019, 15: 102608.; Powder Technol, 2015, 270: 92-97.; J Mol Liq, 2019, 273: 33-36.
There are some English language errors, which should be improved carefully.

Author Response

Q1) The abstract and conclusion should be improved and more concise, and the scientific principles should be further concluded in this work.

A1) The abstract and conclusions were reduced to be more concise and summarized the scientific principles and main achievements of the study.

Q2) The authors should summarize the related attempts on the development of composite materials containing natural fillers with improved performance. It would be necessary to discuss by referring some articles, including: Compos Part A-Appl S, 2019, 124:105472.; Environ Sci Technol, 2019, 53: 6989-6996; J Manuf Process, 2019,45: 520-531.; Appl Clay Sci,2016, 132-133: 175-181.; Results Phys, 2019, 15: 102608.; Powder Technol, 2015, 270: 92-97.; J Mol Liq, 2019, 273: 33-36.

A1) These studies were cited and commented in the Introduction. Please see new references 9, 10, 13,16, 41, and 42.

Q3) There are some English language errors, which should be improved carefully.

A3) English grammar was checked and some expressions and minor mistakes were corrected.

Reviewer 2 Report

In the paper, studies on improving the performance of polylactide reinforced with short flaxseed fibers using different compatibilization strategies are presented.

The paper is properly written from the point of view of material studies. However, each compounding process is determined not only by the material, but also by the compounding operating conditions and screw/die geometry. Compounding is determined by the thermomechanical flow field of the process. which is dependent on the screw/die geometry, operating conditions and material properties.

In the paper, there is an obvious lack of the compounding process characteristics, e.g. screws configuration, flow rate, shear stress, shear strain, residence time etc.

In summary, extrusion as well as injection molding process are not properly described, and the discussion of results should be substantially extended in this respect.

Author Response

Q1) The paper is properly written from the point of view of material studies. However, each compounding process is determined not only by the material, but also by the compounding operating conditions and screw/die geometry. Compounding is determined by the thermomechanical flow field of the process. which is dependent on the screw/die geometry, operating conditions and material properties. In the paper, there is an obvious lack of the compounding process characteristics, e.g. screws configuration, flow rate, shear stress, shear strain, residence time etc. In summary, extrusion as well as injection molding process are not properly described, and the discussion of results should be substantially extended in this respect.

A1) More details about the extruder and its screw configuration were included in Section 2.3, in which a new Figure 2 showing the extruder and the screw configuration was added. More details of the injection molding process were also added in Section 2.4.

Reviewer 3 Report

Dear Authors,

the manuscript entitled "Evaluation of Different Compatibilization Strategies to Improve the Performance of Injection-Molded Pieces of Polylactide Reinforced with Short Flaxseed Fibers” by Ángel Agüero, David Garcia-Sanoguera, Diego Lascano, Sandra Rojas-Lema, J. Ivorra Martinez, Octavio Fenollar and Sergio Torres-Giner investigates green composites made of polylactide and alkali-pretreated flaxseed fibers at 20 wt%. They also investigate different compatibilization strategies based on specific coupling methods:  a multi-functional epoxidized styrene-acrylic oligomer, a random copolymer of poly(styrene-co-glycidyl methacrylate) (PS-co-GMA), and maleinized linseed oil.

The work is generally well thought out, structured and well documented. Some minor things can be improved or completed.

  • In the introduction part as well as in the text, in my opinion, there are some articles that could be mentioned and compared as they concern the use of PLA with natural fibers as filler and the use of the same compatibilizer/additive used in this text, which can also help to explain some behaviors that are verified in this article [1] [2].

  • In the mixing phase it is described how the residence time is about one minute. Have you done any test to prove that the additives that are believed to interact by making a reaction actually manage to do that in that time and at that temperature?

  • Figure 2 have to be adjusted and better organized. It is always better to compare images taken with the same instrument and at the same magnifications. It could be useful to put also a high magnification image of the surface morphology of the three fibers.

  • I ask to pay particular attention to the choice of decimal places to be reported in all the tables and in the text. They must be chosen based on the uncertainty of the measurement. E.g. the elastic modulus with a decimal place.

  • The authors rightly took PLA/FF as an internal comparison as they focus on the various treatments that the fibers undergo. I don't understand the choice of PLA grade. The PLA selected is “for extrusion into mechanically drawn staple fibers or continuous filament, using conventional fiber spinning and drawing equipment”. In particular, the data obtained for the neat PLA have an unusually low modulus for PLA, and a very high impact toughness. Some more comment is needed.

  • Were the fibers dried before melting with PLA? Has the residual humidity of these fibers been checked? Moreover, the fiber treatments can tremendously influence their ability to absorb moisture before being processed. The variations in thermal characteristics (and also the mechanical ones) could be due to that. In addition to the fact that also the alkylation treatment could leave residues that cause a decrease in the molecular weight of the PLA during the process phase. What do you think? I doubt that the adhesive interaction manages to change the degradation temperature.

  • The percentage of crystallinity is calculated on the second heating DSC cycle that is good for general behaviors on the material, but the one calculated on the first DSC cycle on the final products subjected to the mechanical properties could give further information. If they are useful, I recommend adding them.

  • “The degree of crystallinity of the green composite pieces slightly increased when processed with ESAO” quite true, it seems to be the same.

  • Figure 5 can be improved for a better reading and distinction between the various curves.

  • The paragraph “Thermomechanical Properties of the Green Composite Pieces” should be better supported for what is the usefulness is in the article. The explanation of why with this technique it is possible to have such a high result with the GPTMS sample that it does not result from the E modulus at the tensile test and to correlate this result to a better adhesion is to be proved. The data obtained at 75°C have a statistical variation so large that it does not allow in-depth comments that I would suggest to remove from the text. All the other comments can be deduced from evidence already produced in the text. However, the decision is up to the authors.

  • I ask the authors to be more adherent to the real achievements obtained in the text in the conclusions and comments. No chemical reaction have been verified in the reported system in the present article, only supposed. Interfacial adhesion is a very complicated aspect and it is difficult to correlate with only FESEM images that give a subjective evaluation and mechanical tests that can be influenced by many factors (not all examined here e.g. the molecular weight of the matrix or the variation in the properties of the fibers if treated, etc.). Moreover, many times the ability to transfer the stresses between the matrix and the filler is used as a term for the adhesion [3] (thus the strength of the material) not the modulus. I would therefore be careful to define the highest interfacial adhesion sample GPTMS in the text but I suggest to specify better what the author saw and where. Finally, also the part concerning ESAO must be carefully evaluated. The additive was not food contact approved therefore writing food application is not correct and if cross-linking reactions occur its biodegradability has to be confirmed.

[1] Battegazzore D, Frache A, Abt T, Maspoch ML. Epoxy coupling agent for PLA and PHB copolymer-based cotton fabric bio-composites. Compos Part B-Eng. 2018;148:188-97.

[2] Battegazzore D, Abt T, Maspoch ML, Frache A. Multilayer cotton fabric bio-composites based on PLA and PHB copolymer for industrial load carrying applications. Compos Part B-Eng. 2019;163:761-8.

[3] Móczó J, Pukánszky B. Polymer micro and nanocomposites: structure, interactions, properties. Journal of Industrial and Engineering Chemistry. 2008;14(5):535-63.

Author Response

Q1) In the introduction part as well as in the text, in my opinion, there are some articles that could be mentioned and compared as they concern the use of PLA with natural fibers as filler and the use of the same compatibilizer/additive used in this text, which can also help to explain some behaviors that are verified in this article [1] [2].

A1) The studies of Battegazzore et al. were cited and commented in the Results Section 3.2., describing the effect of ESAO on the performance of the green composites. Please see new references 76 and 77.

Q2) In the mixing phase it is described how the residence time is about one minute. Have you done any test to prove that the additives that are believed to interact by making a reaction actually manage to do that in that time and at that temperature?

A2) The residence time was estimated by measuring the time that a blue masterbatch pellet takes from being introduced directly into the extruder until exiting from the die. This information was added in Section 2.3. These compatibilizers were tested individually in previous studies dealing with other condensation polymers or polymer composites so that their interactions have been already proved.

Q3) Figure 2 have to be adjusted and better organized. It is always better to compare images taken with the same instrument and at the same magnifications. It could be useful to put also a high magnification image of the surface morphology of the three fibers.

A3) Two more images at higher magnification and taken at the same magnification were added in the figure (now Figure 3).

Q4) I ask to pay particular attention to the choice of decimal places to be reported in all the tables and in the text. They must be chosen based on the uncertainty of the measurement. E.g. the elastic modulus with a decimal place.

A4) Decimal places were adjusted and corrected in the tables and the text based on the uncertainty of each measurement. In the case of the elastic modulus we consider that is should be based on a decimal place.

Q5) The authors rightly took PLA/FF as an internal comparison as they focus on the various treatments that the fibers undergo. I don't understand the choice of PLA grade. The PLA selected is “for extrusion into mechanically drawn staple fibers or continuous filament, using conventional fiber spinning and drawing equipment”. In particular, the data obtained for the neat PLA have an unusually low modulus for PLA, and a very high impact toughness. Some more comment is needed.

A5) It is correct that the 6201D PLA grade has been particularly designed by the manufacturer for fiber spinning but it was selected for preparing the green composites due to its high fluidity. Though it can affect the optical properties, it is still processable by injection molding applications since it shows a MFI of 15-30 g/10 min (210 °C and 2.16 kg), which is in the range of to those grades for injection molding such as 3001D (MFR of 22 g/10 min), 3051D (15-30 g/10 min) or 3052D (MFI of 15 g/10 min). This was stated in the Materials Section 2.1 and also supported by the references of our previous research studies in which we develop injection-molded PLA pieces using this grade.

Q6) Were the fibers dried before melting with PLA? Has the residual humidity of these fibers been checked? Moreover, the fiber treatments can tremendously influence their ability to absorb moisture before being processed. The variations in thermal characteristics (and also the mechanical ones) could be due to that. In addition to the fact that also the alkylation treatment could leave residues that cause a decrease in the molecular weight of the PLA during the process phase. What do you think? I doubt that the adhesive interaction manages to change the degradation temperature.

A6) As explained in Section 2.2, the alkali-pretreated fibers and silanized fibers were dried at 60 ⁰C for 24 h and 50 ⁰C for 2 days, respectively, prior to be melt processed in order to prevent/reduce hydrolysis. Residual moisture/solvent was determined at ~3.5% by the TGA measurement (please see further details Section 3.4 and Figure 6).

Q7) The percentage of crystallinity is calculated on the second heating DSC cycle that is good for general behaviors on the material, but the one calculated on the first DSC cycle on the final products subjected to the mechanical properties could give further information. If they are useful, I recommend adding them.

A7) It is correct that the second heating provides information of the material since it removes the thermal history, while the first one is more related to the conditions in which the material was cooled. As suggested by the reviewer, to study the crystalline of PLA in the injection-molded pieces, the thermal values corresponding to the first heating were used instead of the second one. These changes were reflected in Section 3.4 and also in the Experimental part, Section 2.5.3.

Q8) “The degree of crystallinity of the green composite pieces slightly increased when processed with ESAO” quite true, it seems to be the same.

A8) The degree of crystallinity of PLA in the green composite increased when it was processed with ESAO. This was property explained in Section 3.4 and it was also confirmed in the measurements performed for the first DSC heating step.

Q9) Figure 5 can be improved for a better reading and distinction between the various curves.

A9) In order to improve the figure (now Figure 6) for a better reading and distinction between the curves, a zoomed area of the onset degradation region (beginning of thermal decomposition) was included.

Q10) The paragraph “Thermomechanical Properties of the Green Composite Pieces” should be better supported for what is the usefulness is in the article. The explanation of why with this technique it is possible to have such a high result with the GPTMS sample that it does not result from the E modulus at the tensile test and to correlate this result to a better adhesion is to be proved. The data obtained at 75°C have a statistical variation so large that it does not allow in-depth comments that I would suggest to remove from the text. All the other comments can be deduced from evidence already produced in the text. However, the decision is up to the authors.

A10) There was a mistake in some of the standard deviation values. The difference between the DMTA and tensile tests results can also be related to the type of analysis. This was described in Section 3.5.

Q11) I ask the authors to be more adherent to the real achievements obtained in the text in the conclusions and comments. No chemical reaction have been verified in the reported system in the present article, only supposed. Interfacial adhesion is a very complicated aspect and it is difficult to correlate with only FESEM images that give a subjective evaluation and mechanical tests that can be influenced by many factors (not all examined here e.g. the molecular weight of the matrix or the variation in the properties of the fibers if treated, etc.). Moreover, many times the ability to transfer the stresses between the matrix and the filler is used as a term for the adhesion [3] (thus the strength of the material) not the modulus. I would therefore be careful to define the highest interfacial adhesion sample GPTMS in the text but I suggest to specify better what the author saw and where. Finally, also the part concerning ESAO must be carefully evaluated. The additive was not food contact approved therefore writing food application is not correct and if cross-linking reactions occur its biodegradability has to be confirmed.

A11) The chemical reactions of these additives have been already described in previous works for PLA or other polyesters or polyamides, which were discussed and supported by references in the whole manuscript. We have also included the statement (please see Section 3.3. and reference 80) in the discussion, which are useful to understand the differences attained in the interfacial adhesion and the mechanical performance. Finally, I would like to indicate that ESAO (Joncryl 4368C) has the food contact approval by its manufacturer, BASF, who indicates that it can be used as long as the final article complies with current migration legislations and this is typically achieved by the use of low contents, in the range of 0.1–1 wt%, which also avoid gel formation.

Round 2

Reviewer 1 Report

This version is much better. There are still some minor English language errors, which should be polished.